# New Histoprognostic Factors to Consider for the Staging of Colon Cancers: Tumor Deposits, Invasive Tumor Infiltration and High-Grade Budding

**DOI:** 10.3390/ijms24043573

**Published:** 2023-02-10

**Authors:** Marc Riffet, Benoît Dupont, Maxime Faisant, Damiano Cerasuolo, Benjamin Menahem, Arnaud Alves, Fatémeh Dubois, Guénaëlle Levallet, Céline Bazille

**Affiliations:** 1Department of Pathology, CHU de Caen, 14000 Caen, France; 2Department of Gastroenterology, CHU de Caen, 14000 Caen, France; 3ANTICIPE, INSERM UMR 1086, UNICAEN, Normandie Université, 14076 Caen, France; 4Biostatistics and Clinical Research Unit, CHU de Caen, 14000 Caen, France; 5Department of Digestive Surgery, CHU de Caen, 14000 Caen, France; 6ISTCT, GIP CYCERON, CNRS, UNICAEN, Normandie Université, 14074 Caen, France; 7Structure Fédérative D’oncogénétique cyto-Moléculaire, CHU de Caen, 14000 Caen, France

**Keywords:** colorectal cancer, histoprognostic factors, budding, invasion front, isolated tumor deposits

## Abstract

Colorectal cancer is a major public health issue due to its high incidence and mortality. It is, therefore, essential to identify histological markers for prognostic purposes and to optimize the therapeutic management of patients. The main objective of our study was to analyze the impact of new histoprognostic factors, such as tumor deposits, budding, poorly differentiated clusters, mode of infiltration, the intensity of inflammatory infiltrate and the type of tumor stroma, on the survival of patients with colon cancer. Two hundred and twenty-nine resected colon cancers were fully histologically reviewed, and survival and recurrence data were collected. Survival was analyzed using Kaplan–Meier curves. A univariate and multivariate Cox model was constructed to identify prognostic factors for overall survival and recurrence-free survival. The median overall survival of the patients was 60.2 months and the median recurrence-free survival was 46.9 months. Overall survival and recurrence-free survival were significantly worse in the presence of isolated tumor deposits (log rank = 0.003 and 0.001, respectively) and for an infiltrative type of tumor invasion (log rank = 0.008 and 0.02, respectively). High-grade budding was associated with a poor prognosis, with no significant difference. We did not find a significant prognostic impact of the presence of poorly differentiated clusters, the intensity of the inflammatory infiltrate or the stromal type. In conclusion, the analysis of these recent histoprognostic factors, such as tumor deposits, mode of infiltration, and budding, could be integrated into the results of pathological reports of colon cancers. Thus, the therapeutic management of patients could be adjusted by providing more aggressive treatments in the presence of some of these factors.

## 1. Introduction

Colorectal cancer (CRC) is a major public health issue due to its high incidence and mortality. Well-defined histoprognostic factors allow adaptation of the therapeutic management of patients according to the evolutionary stage of the disease. The main factor is the pTNM stage, which allows us to evaluate the locoregional and distant extension of the disease. Other histoprognostic factors are recognized, such as the histological type, the grade of differentiation, and the presence of lymphatic or venous tumor emboli. The majority of CRCs correspond to adenocarcinomas not otherwise specified; however, some histological types, such as independent cell adenocarcinoma, micropapillary adenocarcinoma, adenosquamous carcinoma, or carcinoma with a sarcomatoid component, have a poor prognosis [1,2,3,4,5,6], whereas adenoma-like adenocarcinoma has a better prognosis [7]. High-grade (or poorly differentiated) differentiation, the presence of tumor emboli, and the presence of perineural involvement are factors suggestive of a poor prognosis [8,9,10].

New histoprognostic factors, which are not yet required in pathology reports, are being evaluated. Among the histoprognostic factors investigated more recently, tumor deposits correspond to tumor clusters at a distance from the tumor. In the eighth edition of the UICC classification, the presence of tumor deposits without associated lymph node metastasis is classified as N1c and is not reported if lymph node metastases are present. However, recent studies showed that tumor deposits might constitute a prognostic factor independent of the presence or absence of lymph node metastasis [11,12,13]. Furthermore, some studies have shown that high-grade budding has a negative impact on patient survival [14,15,16]. For pT1 tumors, the presence of budding correlates with a higher risk of lymph node metastasis [17]. The presence of poorly differentiated clusters (PDCs) is also a factor indicating a poor prognosis in the pT1 stages, as it correlates with a higher risk of lymph node metastasis [18,19,20]. Other histoprognostic factors are emerging in the literature, such as the type of tumor infiltration, the type of stroma, and the intensity of the inflammatory infiltrate.

The main objective of this work was to study the impact of these new histoprognostic factors (tumor deposits, budding, poorly differentiated clusters, mode of infiltration, intensity of inflammatory infiltrate, and type of tumor stroma) on overall survival and recurrence-free survival, based on a retrospective cohort of patients with colon cancer operated on at the University Hospital of Caen, with variable locations and stages. This study will allow pathologists to identify the histoprognostic factors to recognize and record them in the histological report for better therapeutic management and, in particular, to support decision making about adjuvant chemotherapy.

## 2. Results

### 2.1. Clinical Population Characteristics

The clinical criteria are detailed in Table 1. The average age of the cohort was 71 years, with extremes ranging from 27 to 99 years. One hundred and twenty patients were men, representing 52.4% of the study cohort. The tumor was located on the right in 52.0% of cases and on the left in 48.0% of cases. Stage I patients constituted 9.1% of the patients in the cohort, 35.4% were stage II patients, 38.0% were stage III patients, and 17.5% were stage IV patients. Among the 229 patients included in this study, no patient was lost to follow-up. The mean follow-up of patients was 51.2 months (0–103) with a median of 58.43 months. During the study, 96 patients died. One hundred and eleven patients were alive after 5 years, providing an overall 5-year survival rate of 48.5%. The patients had a median overall survival of 60.2 months and a median recurrence-free survival of 46.9 months.

### 2.2. Histoprognostic Factors

The histological and molecular criteria are detailed in Figure 1 and Table 2. Of the 229 colic adenocarcinomas analyzed, most were low grade: well differentiated (44.1%). Twenty-nine tumors were mucinous adenocarcinoma (12.7%). In the majority of cases (61.1%), the tumors were stage pT3. The pT1 and pT2 stages represented 3.0 and 7.8%, respectively. Fifty-three tumors (22.2%) ulcerated the colic serosa (stage pT4a), and two (0.9%) invaded adjacent organs (stage pT4b). One hundred and thirteen patients (49.3%) had no lymph node metastases. We observed the presence of tumor emboli in 96 patients (41.9%).

Nine patients (3.9%) had only isolated tumor deposits with no formally identifiable lymph node metastasis (N1c), and fifty-three patients had both tumor deposits and lymph node metastasis. Concerning the evaluation of budding and poorly differentiated clusters, a large majority of the tumors had a low grade of budding or PDC (76.0% and 74.2%, respectively). Eighteen (7.8%) and thirty-two (14%) patients had budding or high-grade PDC, respectively. For two patients, budding and low-grade clusters could not be determined, as these patients had adenocarcinoma with an independent cell component. Analysis of the intra- and peritumoral inflammatory infiltrate showed that the intensity of the infiltrate was often moderate (45.89%). For six tumors (2.6%), an intense Crohn-like inflammatory infiltrate was observed. Concerning the type of tumor infiltration on the front, it was of the infiltrative type in 151 cases (65.9%). The tumor stroma was predominantly of intermediate type (71.6% of cases). Only one tumor (0.4%) had an immature stroma, but the evaluation of the type of stroma remains subjective. Microsatellite phenotype analysis by immunohistochemistry was performed in 14.8% of cases with an MSI phenotype for 10 tumors. Thirty-four patients (42.5% of cases analyzed) had a KRAS or NRAS mutation, and six had a BRAF mutation (8% of cases analyzed).

The median overall survival of patients was 60.2 months, and the median recurrence-free survival was 46.9 months. According to the log rank test, we observed significantly worse overall survival and recurrence-free survival in patients with colon cancer with tumor deposits (*p* < 0.003 and 0.001, respectively) (Figure 2 and Appendix A, Table 3). We also observed worse overall survival and recurrence-free survival when the tumor presented an infiltrative mode of invasion (*p* < 0.008 and 0.02, respectively) compared to an expansive mode (Figure 2 and Appendix A). Concerning the analysis of the impact of budding, we observed a tendency toward worse overall survival and recurrence-free survival for grade 2 budding and even more so for grade 3 budding; however, the difference was not significant (*p* = 0.43 and 0.11, respectively) (Appendix A and Table 3). We did not find any significant difference in the other histoprognostic factors, such as PDC, the intensity of inflammatory infiltrate, or the stromal type.

Clinical and histologic factors associated with overall and recurrence-free survival are presented in Table 4 according to a Cox model. Factors independently associated with overall survival with multivariate analysis were male sex (*p* = 0.003), age less than 70 years (*p* < 0.0001), right location (*p* = 0.01), absence of tumor deposits (*p* = 0.003), and Stage III and IV compared to Stage I/II (*p* < 0.0001). Factors independently associated with recurrence-free survival were male sex (*p* = 0.0002), age <70 years (*p* = 0.006), absence of tumor deposits (*p* = 0.005), and stage III and IV (*p* < 0.0001). Tumor invasion in an infiltrating mode tended to result in lower overall survival and recurrence-free survival rates but was not significant in multivariate analysis.

## 3. Discussion

In our study, we included all clinical stages of patients operated on for colon cancer. We showed that clinical factors independently associated with overall survival, including male sex, age less than 70 years, right location, absence of synchronous metastases and absence of tumor deposits, infiltration mode and budding, were also important factors to consider.

In our cohort, the average age of the patients was 71 years, which is consistent with data from the French National Cancer Institute which describe an average age of 71 years at diagnosis in men and 73 years in women [21]. On a global scale, studies show an average age of between 53 and 73 years, depending on sex [22,23,24,25]. However, in recent years, there has been a decrease in the incidence in patients over 50 years of age and an increase in patients under 50 years of age. This shift in incidence is thought to be related to several factors, such as the lack of screening in patients under 50 years of age, or environmental factors, such as dietary habits, sedentary lifestyle, and smoking, particularly in developing countries [23,24,25,26].

We observed a slight predominance of right colon tumors in our cohort. Depending on the study, this right/left location predominance varies, but most often, left colon cancers predominate [27,28,29]. Nevertheless, although colon cancers were historically more frequently located on the left, the incidence of tumors located in the right colon has increased over the years [28]. Concerning the clinical stages at diagnosis, the incidence is also variable according to the studies, but colon cancers are mainly diagnosed in stages II or III [24,29,30]. We made the same observation in our cohort. According to these data, our cohort thus appears homogeneous. However, we observed a lower overall survival at 5 years, at 48.5%, in our cohort, whereas in the literature, the overall survival at 5 years is reported to be between 63.5% and 71% [24,29,31].

The main objective of our study was to analyze the impact of different histoprognostic factors on the 5-year overall and recurrence-free survival of patients in our cohort. Our results show that the presence of isolated tumor deposits is a factor indicative of a poor prognosis. The study conducted by Nagtegaal et al. highlights the negative impact of tumor deposits in CRC on uni- and multivariate overall survival (HR 2.9, HRa 2.2) and on recurrence-free survival (HR 2.2 and HRa 2.0) [11]. Similar results were observed in the meta-analysis by Lord et al. on overall (HR 1.63) and recurrence-free (HR 1.77) survival [32]. On the other hand, the presence of tumor deposits associated with lymph node metastases has a worse prognosis than tumor deposits alone or lymph node metastases alone, as observed by Mirkin et al. [12]. They demonstrated a higher survival for tumor deposits alone (HR 0.56) or lymph node metastases alone (HR 0.64). Therefore, it has been proposed to include a new category in the “N” status of the pTNM classification for tumor deposits regardless of lymph node metastasis in its next revision [11,12,13].

Regarding the influence of the type of invasion on the tumor invasion front, we observed for the infiltrative mode of invasion a lower 5-year overall survival and recurrence-free survival (*p* = 0.008 and 0.02, respectively) and that this mode tended to have a poorer prognosis (but in a nonsignificant way for multivariate analysis) of both the overall survival (HR 1.65, *p* = 0.009) and recurrence-free survival (HR 1.51, *p* = 0.023). Similarly, Qwaider et al. observed a pejorative impact of the infiltrative mode on the survival of patients with stage II or III CRC, but they did not demonstrate a significant difference in HRs for overall survival and recurrence-free survival [33]. On the other hand, Li et al. observed, in patients with stage I to III CRC, a better prognosis when the mode of infiltration was expansive for overall survival (HR 0.73) and recurrence-free survival (HR 0.722) [34]. In addition, Karamitopoulou et al. observed that the presence of a more than 90% infiltrative component was associated with a poor prognosis for stages II and III [35]. It has also been reported that the infiltrative mode of invasion is more frequently associated with a more advanced pTNM stage, poor tumor differentiation, a higher budding grade, and the presence of lymphatic and vascular tumor emboli [34,35]. Molecularly, the expansive mode of invasion is frequently associated with CRCs with a MSI phenotype, whereas the infiltrative mode of invasion is frequently associated with a BRAF gene mutation [34].

Patients with high-grade budding tend to have lower survival rates; however, the impact appears to be nonsignificant for 5-year overall and recurrence-free survival (log rank = 0.43 and 0.11, respectively). Numerous studies have described the prognostic impact of the presence and grade of budding in CRC. These studies have distinguished between low-grade budding (0 to 9 isolated cells) and high-grade budding (≥10 isolated cells). The results showed that stage II CRC patients with high-grade budding had worse 5-year survival: 89.0% for low-grade budding compared with 66.7% for high-grade budding [36,37,38]. These studies also highlighted that high-grade budding had a poor prognosis, with an HR of 2.57 to 4.89 [36,37,38,39,40]. Compared with our cohort, most of these studies included patients with stage II CRC, whereas we included patients with CRC regardless of stage. Therefore, it would be interesting in our study to analyze the impact of budding on the survival of stage II patients. Furthermore, Graham et al. have shown that high-grade budding is more frequently associated with CRC of the MSS phenotype and with the presence of a KRAS mutation [14,15,16,17,39]. In addition, more frequent KRAS and BRAF gene alterations are also observed in CRCs with high-grade PDC [14,18,19,20]. The budding phenomenon is characterized by an epithelial–mesenchymal transition phenotype and is considered an important step in metastasis formation.

Regarding the intensity of the inflammatory infiltrate, we did not find any significant difference in overall survival or recurrence-free survival. However, the intensity of the lymphocytic inflammatory infiltrate in response to tumor invasion is frequently described as an important prognostic factor. Indeed, patients have better overall survival when there is a large lymphocytic inflammatory infiltrate [41,42]. The study conducted by Haruki et al. sought to characterize the impact of this inflammatory infiltrate on the survival of CRC patients by looking not only at the intensity of the infiltrate, but also at its location. They defined the intensity of inflammation in a semiquantitative manner (rated 0 (none), 1+ (low), 2+ and 3+ (high)) and determined the location of the inflammatory infiltrate according to four components: transmural inflammatory infiltrate (or Crohn-like lymphocytic reaction), peritumoral, periglandular intratumoral, and tumor-infiltrating lymphocytes (TILs). Their results showed that patients have better overall survival when the inflammatory infiltrate is moderate or intense (2+ to 3+), whether transmural, peritumoral, or periglandular intratumoral. A 2+ or 3+ score for TILs also correlates with better patient survival [21]. In our study, the semiquantitative assessment probably lacked power compared with an accurate quantitative analysis.

## 4. Materials and Methods

### 4.1. Study Population

We performed a monocentric retrospective cohort study including all patients who underwent surgery for colon cancer at Caen University Hospital from January 2010 to September 2013. A total of 229 patients were included, regardless of disease stage. Rectal cancer patients were excluded due to the performance of neoadjuvant radiotherapy, which leads to histological changes.

### 4.2. Clinical Data

The clinical data of each patient were collected using the software of the information system of the University Hospital or from clinical files. This information included the date of surgery, the right or left colon location of the tumor, its occlusive or nonocclusive character, the presence of synchronous hepatic or extrahepatic metastasis, and the occurrence of a recurrence, its date of discovery and any possible treatment. Finally, the dates of the last follow-up and the date of death, if applicable, were sometimes retrieved after telephone contact to establish overall survival and recurrence-free data.

### 4.3. Morphological Criteria

Morphological criteria were collected on the macroscopic report. All histological criteria were analyzed and reviewed by two pathologists (CB and MR) based on the totality of the histological slides stained with hematoxylin eosin saffron: tumor size in centimeters; grade of differentiation (well differentiated, moderately differentiated, poorly differentiated, mucinous variant); pTNM stage according to the 2017 UICC classification (8th edition); and the presence or absence of lymphatic and/or venous tumor emboli.

The presence or absence of tumor deposits was assessed. Tumor deposits are located in the pericolic fat, regardless of their size. They lack continuity with the primary tumor and differ from metastatic lymph node invasion by the absence of identifiable lymph node parenchyma and from emboli by the absence of a vascular wall. Tumor budding is defined by the presence of isolated cells or clusters of up to 4 tumor cells. The search for budding should be performed according to the recommendations published by Lugli et al. about the invasion front of tumor proliferation [14]. Budding was characterized on the tumor invasion front and classified into 3 grades. The field with the highest amount of budding on the invasion front was selected, and then the number of single cells or clusters of up to 4 cells was scored at ×200 magnification on an area of 0.785 mm^2^. Budding was graded as grade 1 (low) between 0 and 4 isolated cells or clusters, grade 2 (intermediate) for 5 to 9 isolated cells or clusters, or grade 3 (high) for at least 10 isolated cells or clusters (14). PDCs are characterized on the invasion front by tumor clusters of at least 5 cells without glandular structure formation. The determination of PDC grades is the same as the technique used to grade budding. PDCs scored on the tumor invasion front were classified into 3 grades: low, intermediate, or high. CRCs invade the colon wall in two ways: in an expansive or pushing mode, characterized by a well-limited border between the tumor structures and the nontumor tissue, or in an infiltrative mode, when a clear boundary between the tumor cells and the adjacent nontumor tissue was not defined [33,35]. The intensity of the intratumoral and peritumoral inflammatory infiltrate was semiquantitatively categorized into 4 grades: low, moderate, marked, and Crohn-like with germ-center lymphoid follicle formation. We also characterized the invasion front: well delineated, pushing type or poorly delineated, infiltrative type, as well as the type of stroma, defined as mature (fibrous), intermediate (fibrous with the presence of keloid fibers) or immature (myxoid).

### 4.4. Standard HES Staining

Surgical specimens, which were received fresh, were placed in buffered formalin for 24 to 48 h. The duration was determined by the size of the specimen. Once fixed, the tumor was sampled and the mesocolon lymph nodes were retrieved. The samples followed the usual circuit of fixing and paraffin embedding of specimens. Briefly, the specimens were dehydrated in an alcohol bath to increasing degrees and then in a xylene bath and then dipped in liquid paraffin. After paraffin embedding, 3 μm sections were cut with a microtome and placed on a glass slide. The slides were then deparaffinized and stained with HES (hematoxylin, eosin, saffron) in an automated stainer.

### 4.5. Statistical Analysis

Qualitative and quantitative variables were compared using a Chi2 or Student’s *t*-test, respectively (Mann–Whitney test or Fisher’s exact test when the validity conditions of the Student’s and Chi2 tests were not verified). Survival was analyzed using Kaplan–Meier curves. A univariate and multivariate Cox model was performed to identify prognostic factors for overall survival and recurrence-free survival. The variables that were associated with the variable to be explained (survival) were selected by a backward selection process to perform the final multivariate model. A difference was considered significant when *p* was less than 0.05. All statistical analyses were performed using SAS 9.4 software (SAS Institute, Cary, NC, USA).

## 5. Conclusions

The identification of new prognostic markers for the management of colon cancer is a major issue. Our work shows a negative impact on survival when there are isolated tumor deposits and when the tumor invasion mode is infiltrative. We also showed a nonsignificant trend toward a worse prognosis when high-grade budding was present. These results highlight the value of systematically identifying these new histoprognostic factors in histological reports, which could allow us to adjust the therapeutic management of patients by providing more aggressive treatments in the presence of some of these factors. The prognostic performances of these three new histological factors have to be assessed in an independent cohort.

## Figures and Tables

**Figure 1 ijms-24-03573-f001:**
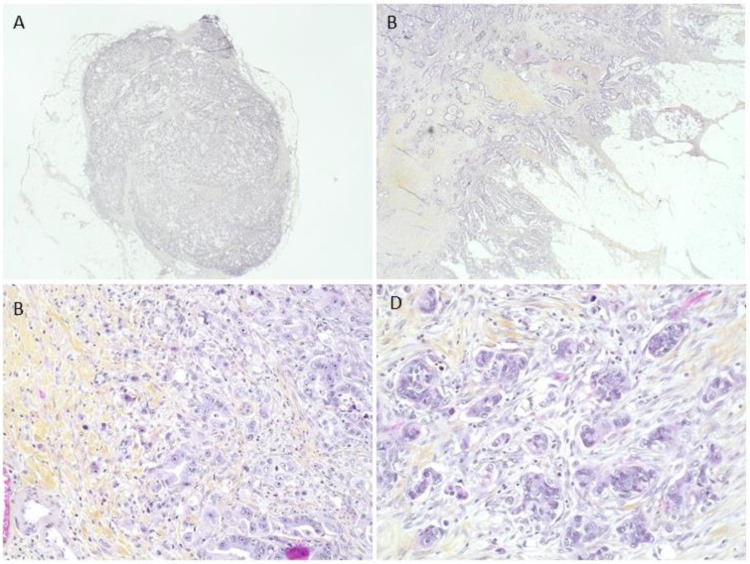
**Prognostic histological factors studied.** (**A**) Tumor deposit without identifiable lymph node parenchyma or vascular wall. (**B**) Tumor front invasion in an infiltrative mode. (**C**) High-grade budding for at least 10 isolated cells or clusters. (**D**) Poorly differentiated clusters with tumor clusters of at least 5 cells without glandular structure formation.

**Figure 2 ijms-24-03573-f002:**
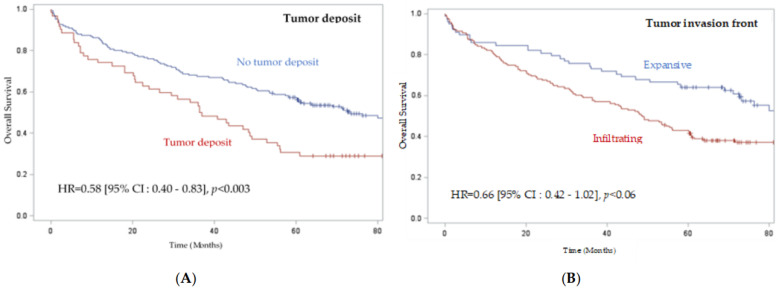
**Prognostic factors studied and overall survival in 229 patients with colon cancer.** (**A**) Tumor deposit impact on overall survival in patients with colon cancer. (**B**) Impact of tumor front invasion on overall survival in patients with colon cancer.

**Table 1 ijms-24-03573-t001:** Clinical characteristics of the cohort.

Clinical Characteristics	All Patients N = 229	
	N	%
Age (years old)	71 (27–99)	
Sex (female/male)	109/120	47.6/52.4
Location (right/left)	119/110	52.0/48.0
Tumor size (cm)	5.05 (0.4–16)	
Clinical stage	21/81/87/40	9.1/35.4/38.0/17.5
I/II/III/IV		
Intestinal occlusion	67	29.2
Synchronous metastasis (No/hepatic/extrahepatic)	188/35/6	82.1/15.3/2.6
No recurrence	180	78.6
Recurrence local/hepatic/extrahepatic	18/19/8	7.9/8.3/3.5
Death	96	40.6

**Table 2 ijms-24-03573-t002:** Histological characteristics of the cohort.

Histological Characteristics	All Patients N = 229	
	N	%
Differentiation (well/moderate/poor) Mucinous	101/76/23 29	44.1/33.2/10.0 12.7
Stage T 1/2/3/4a/4b	7/18/140/51/2	3.0/7.8/61.1/22.2/0.9
Stage N 0/1a/1b/1c/2	113/34/37/9/36	49.3/14.9/16.2/3.9/15.7
Emboli (no, lymphatic, venous)	133/51/45	58.1/22.3/19.6
Tumor deposit	62	27.1
Budding (1/2/3/NA ^1^)	174/35/18/2	76.0/15.3/7.8/0.9
CPD (1/2/3/NA ^1^)	170/25/32/2	74.2/10.9/14.0/0.9
Tumor invasion front (expansive/infiltrating)	78/151	34.1/65.9
Inflammation (low/moderate/intense/Crohn-like)	62/105/56/6	27.1/45.8/24.5/2.6
MSI (MSS/MSI/NA ^1^)	24/10/195	10.5/4.4/85.1
RAS (no, mutated/NA ^1^)	46/34/149	20.1/14.8/65.1
BRAF (no, mutated/NA ^1^)	69/6/154	30.1/2.6/67.3

^1^ Not available.

**Table 3 ijms-24-03573-t003:** **Results of the log rank test: mean survival time according to prognostic factors studied in 229 patients with colon cancers** for overall (OS) and recurrence-free survival (SSR).

**OS**	**Mean Survival Time (Months)**	***p*-Value**
No Tumor deposit/Tumor deposit	73.13/36.93	0.003
Tumor invasion front: expansive/infiltrating	88.23/48.73	0.008
Budding 1/2/3	71.63/42.65/42.90	0.43
**SSR**	**Mean Survival Time (Months)**	***p*-Value**
No Tumor deposit/Tumor deposit	58.20/22.43	0.001
Tumor invasion front: expansive/infiltrating	73.13/36.93	0.02
Budding 1/2/3	49.8/42.23/14.51	0.11

**Table 4 ijms-24-03573-t004:** Evaluation of the impact of clinical and histological prognostic factors on overall (OS) and recurrence-free survival (SSR) of the patients studied, according to a Cox model. Stages were evaluated without considering N1c staging to keep tumor deposits as an independent value.

Characteristics	HR	*p*	HR ^a^	*p*
**OS**	**Univariate model**		**Multivariate model**	
Sex (male)	1.43	0.04	1.73	0.003
Age < 70 y	1.68	0.004	2.25	0.0001
Right location	1.35	0.92	1.60	0.01
No tumor deposit	0.58	0.003	0.57	0.003
Infiltrating invasion front	1.65	0.009	-	-
Stage III	1.72	0.0001	1.62	0.0001
Stage IV	3.9		4.16	
**SSR**	**Univariate model**		**Multivariate model**	
Sex (male)	1.70	0.002	1.95	0.002
Age < 70 y	1.43	0.04	1.84	0.006
Right location	1.05	0.76	-	-
No tumor deposit	0.57	0.002	0.59	0.005
Infiltrating invasion front	1.51	0.023	-	-
Stage III	1.75	0.0001	1.78	0.0001
Stage IV	4.50		4.16	

HR: Hazard Ratio; HR ^a^: Adjusted Hazard Ratio.

## Data Availability

All data are stored at the Caen University Hospital center and can be made available upon request.

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
