# Peer review of "New Histoprognostic Factors to Consider for the Staging of Colon Cancers: Tumor Deposits, Invasive Tumor Infiltration and High-Grade Budding"

_ijms, 2023, doi:10.3390/ijms24043573_

Round 1
Reviewer 1 Report
Congratulations on an excellent analysis of the current attempt to describe biology better through pathology reports. Interesting to see still a relatively low uptake of molecular markers in routine diagnostics. Overall this is a very sound and well-written manuscript that reflects current practice and objective applications.
Author Response
Thank you for your review and congratulations.
Reviewer 2 Report
Authors present a retrospective study, from patients submitted to surgery due to colon cancer between 2010 and 2013.
The study is important, and interesting for readers.
Some doubts were raised during the reading of the manuscript:
- The invasion mode should not be stated as an important factor for OS in the first paragraph of the discussion, at least in the way it is presented. Besides not being statistically significant according to table 2, authors do not show the median/mean values of survival for this, or any, of the risk factors. I think that a table showing the comparison between the values in months of OS and recurrence-free survival according to the presence or absence of the studied risk factors is mandatory, besides presenting p-values.
- In the last paragraph of the results it is stated: "Tumor invasion in an expansive mode tended to have a better overall survival and recurrence-free survival, but the difference was not significant 209 (p=0.16)." But according to S1, recurrence-free survival was significantly different from infiltrative mode. Please clarify, either in results and discussion sections, because in the latter it is said: "Regarding the influence of the type of invasion on the tumor invasion front, we observed for the infiltrative mode of invasion a lower 5-year overall survival and recurrence-free survival (p=0.008 and 0.02, respectively) and that the expansive mode tended to have a better prognosis, but in a nonsignificant way". Patients had one of the two invasion modes (n=78vs151, total 229), so how is infiltrative mode significantly worse comparing to expansive, but expansive is not significantly better comparing to infiltrative?
- In supplementary material, only recurrence-free survival is showed, not overall survival. Please correct Figure S1 legend.
- In the last paragraph of the results, "age greater than 70 years" is not correct; instead, age <70 years should be mentioned.
- What was the mean/median follow-up of the patients?
- I think it would be useful to subanalyze the risk factors according to the cancer staging. Some pT1 cancers, for example, can be resected by endoscopy, namely endoscopic submucosal dissection. It would be interesting to see if some of the risk factors that can be analyzed in the ESD specimen (as budding or mode of invasion in the tumor front) may help predicting which patients could have a lower recurrence-free survival or OS.
- In table 2, OS, right location was not significant in the univariate analysis; did it enter the multivariate model? Other studied risk factors, as budding, must also be presented.
Reviewer 3 Report
In this retrospective study, the authors analyzed the impact of 16 histological factors on the survival of 229 patients with colon cancer. They identified 3 features, namely tumor deposits, invasive tumor infiltration and high-grade budding, that were associated with patients' survival. Overall, the findings are of potential clinical significance. However, the current study suffers from major pitfalls that need to be addressed before the manuscript can be further considered for publication in IJMS:
1. The authors did not directly include TMN staging in the Cox regression model (although synchronus metastasis was included). It is therefore difficult to assess if the new features could predict prognosis in a TNM-staging independent manner. Moreover, subgroup analysis (e.g. stage I/II patients only or stage III/IV patients only) should also be performed to address this issue.
2. The prognostic performance of these three new histological factors have to be assessed in an independent cohort.
3. MSI is a major prognostic factor. However, the MSI status of 85.1% of recruited colon cancer patients is unknown. It is therefore impossible to know if the identified histological features could predict prognosis independent of MSI.
Round 2
Reviewer 3 Report
The authors have now addressed my previous concerns.